# The Biological and Clinical Consequences of RNA Splicing Factor U2AF1 Mutation in Myeloid Malignancies

**DOI:** 10.3390/cancers14184406

**Published:** 2022-09-10

**Authors:** Yangjing Zhao, Weili Cai, Ye Hua, Xiaochen Yang, Jingdong Zhou

**Affiliations:** 1Jiangsu Key Laboratory of Medical Science and Laboratory Medicine, School of Medicine, Jiangsu University, Zhenjiang 212013, China; 2Institute of Medical Genetics and Reproductive Immunity, School of Medical Science and Laboratory Medicine, Jiangsu College of Nursing, Huai’an 223005, China; 3Institute of Oncology, Affiliated Hospital of Jiangsu University, Jiangsu University, Zhenjiang 212001, China; 4Department of Thyroid and Breast Surgery, Affiliated Kunshan Hospital of Jiangsu University, Kunshan 215300, China; 5Department of Hematology, Affiliated People’s Hospital of Jiangsu University, Zhenjiang 212002, China

**Keywords:** U2AF1, mutation, RNA splicing, prognosis, myelodysplastic syndromes, acute myeloid leukemia, myeloid malignancies

## Abstract

**Simple Summary:**

U2 small nuclear RNA auxiliary factor 1 (U2AF1) is one of the most important RNA splicing genes involved in regulating the alternative splicing of pre-mRNA. U2AF1 mutation is a genetic driver event in the initiation of myelodysplastic syndromes (MDSs) and frequently occurs in myeloid malignancies. U2AF1 mutation can severely impair hematopoiesis, drive tumor progression, adversely affect disease prognosis, and promote leukemic transformation. This review summarizes the biological and clinical implications of the oncogenic role of U2AF1 mutation in myeloid tumors. Our work provides important and comprehensive insights into the development of the *U2AF1* mutation as a novel prognostic biomarker and therapeutic target for myeloid malignancies.

**Abstract:**

Mutations of spliceosome genes have been frequently identified in myeloid malignancies with the large-scale application of advanced sequencing technology. U2 small nuclear RNA auxiliary factor 1 (U2AF1), an essential component of U2AF heterodimer, plays a pivotal role in the pre-mRNA splicing processes to generate functional mRNAs. Over the past few decades, the mutation landscape of U2AF1 (most frequently involved S34 and Q157 hotspots) has been drawn in multiple cancers, particularly in myeloid malignancies. As a recognized early driver of myelodysplastic syndromes (MDSs), U2AF1 mutates most frequently in MDS, followed by acute myeloid leukemia (AML) and myeloproliferative neoplasms (MPNs). Here, for the first time, we summarize the research progress of U2AF1 mutations in myeloid malignancies, including the correlations between U2AF1 mutations with clinical and genetic characteristics, prognosis, and the leukemic transformation of patients. We also summarize the adverse effects of U2AF1 mutations on hematopoietic function, and the alterations in downstream alternative gene splicing and biological pathways, thus providing comprehensive insights into the roles of U2AF1 mutations in the myeloid malignancy pathogenesis. U2AF1 mutations are expected to be potential novel molecular markers for myeloid malignancies, especially for risk stratification, prognosis assessment, and a therapeutic target of MDS patients.

## 1. Introduction

Myeloid neoplasms constitute a group of diverse hematological disorders, including acute myeloid leukemia (AML), myelodysplastic syndromes (MDS), myeloproliferative neoplasms (MPNs), and MDS/MPN [1]. MDS is a group of heterogeneous myeloid neoplasms originating from the clonal proliferation of hematopoietic stem cells (HSCs) in the bone marrow. The clinical features of MDS are ineffective hematopoiesis, peripheral blood cytopenia, recurrent genetic abnormalities, and an increased risk of AML progression [2]. Approximately 30% of MDS patients finally develop with an increased blast count to ≥20% of total nucleated cells in the bone marrow and are diagnosed as “secondary AML” (sAML), which accounts for 25% to 35% of all AML cases [3]. Next-generation sequencing studies have revealed the genetic clonal evolutions of MDS and AML, and identified hundreds of driver mutations associated with the pathogenesis of leukemia. These recurrently mutated genes mainly enrich in RNA splicing, DNA methylation, histone modification, transcriptional regulation, signal transduction, and cohesin complexes [3,4]. Recently, a new risk scoring system (the IPSS-Molecular, IPSS-M) was announced based on the revised International Prognostic Scoring System (IPSS-R) for MDS. The IPSS-M incorporates molecular information and somatic gene mutations into the risk stratification. The prognostic gene panel in the IPSS-M model includes RNA splicing (SF3B1, U2AF1, and SRSF2), DNA methylation (DNMT3A and IDH2), histone modification (EZH2 and ASXL1), transcriptional regulation (TP53, RUNX1, NPM1, and ETV6), signal transduction (FLT3, NRAS, KRAS, and CBL), and the fusion gene (MLL) [5].

Somatic mutations of pre-mRNA splicing factors are common acquired mutations and early genetic events in MDS and related myeloid neoplasms, and also occur in some solid tumors [6]. The spliceosome is a dynamic ribonucleoprotein complex in the eukaryotic cell nucleus assembled by small nuclear RNAs (snRNA) and numerous proteins. The spliceosome catalyzes the splicing reaction to remove introns from pre-mRNA of the initial transcription products and join exons to produce mature mRNAs. Pre-mRNA splicing is a highly coordinated sequential process to allow for the generation of multiple protein isoforms from a single gene and acts as an essential regulator of gene expression [7]. The U2-dependent spliceosome catalyzes the most dominant type of pre-mRNA, intron U2-type introns in eukaryotes, whereas the U12-type introns catalyzed by the U12-dependent spliceosome only represent less than 1% of human introns. The U2-dependent spliceosome is a variety of small nuclear ribonucleoproteins (snRNPs) composed of five kinds of snRNA (U1/2/4/5/6) and a group of proteins [8,9].

Until now, multiple recurrently mutated splicing factors have been found in hematologic malignancies and affect the splicing process, including splicing factor 3b subunit 1 (SF3B1), U2 small nuclear RNA auxiliary factor 1 (U2AF1), serine- and arginine-rich splicing factor 2 (SRSF2), and Zinc finger CCCH-type, RNA binding motif and serine/arginine-rich 2 (ZRSR2) [10]. U2AF1 (also known as U2AF35) is a U2 auxiliary factor that forms a heterodimer with U2AF2 to recognize the 3′ splice site (3′SS) and recruit U2 snRNP. U2AF1 mutations have been identified in hematological malignancies and multiple solid tumors [11,12]. The high mutation frequency of U2AF1 in myeloid neoplasms, especially in MDS and AML, contributes to abnormal hematopoiesis and cancer progression [13]. As one of the most common mutations in myeloid neoplasms and an early oncogenic driver of myelodysplasia, a systematic understanding of the U2AF1 mutation is currently lacking. Thus, here, we comprehensively summarize the genetic landscape, clinical relevance, molecular pathogenesis, and therapeutic applications of oncogenic U2AF1 mutations in myeloid malignancies.

## 2. U2AF1 Participates in Pre-mRNA Splicing Process

### 2.1. RNA Splicing Cycle

Pre-mRNA splicing is an essential step in the post-transcriptional regulation of gene expression. This process aims to remove the noncoding intron sequences from pre-mRNA and ligate the neighboring coding regions (exons) to produce mature mRNAs for protein biosynthesis [14]. The U2 spliceosome-dependent splicing cycle is a highly ordered process with three stages: spliceosome assembly and activation, splicing reaction execution, and spliceosome disassembly [15]. This process initiates with the recruitment of U1 snRNP and splicing factor SRSF6 to the 5′SS. The large subunit of U2AF (U2AF2) and splicing factor 1 (SF1) bind to branch point sequence (BPS) and the polypyrimidine tract (PT), respectively. The small subunit of U2AF (U2AF1) binds to the AG dinucleotide site of the 3′SS and interacts with U2AF2, SRSF2, and ZRSR2 to recognize the 3′SS (E complex). Subsequently, these interactions promote the U2 snRNP complex containing SF3A1, SF3B1, and SF3B3, to replace SF1 and bind to BPS (A complex). Subsequently, the preassembled U4/U6.U5 tri-snRNP complex is recruited by U1 and U2 snRNPs to the BPS and PT regions together with PRFP8 (B complex). The U1 and U4 snRNPs are then released to form a catalytically active complex of the spliceosome and catalyze the first and second transesterification reactions, respectively (B^act^/B*/C complex). They catalyze the excision and release of intron loop/lariat, as well as the ligation of exons, synthesis of mature mRNA, and disassembly of the spliceosome components for recycling (Figure 1) [10,15,16].

### 2.2. Alternative Splicing Patterns

It is generally known that there are two types of pre-mRNA splicing: constitutive splicing and alternative splicing. Constitutive splicing joins all exons in the pre-mRNA, whereas alternative splicing involves or excludes alternative exons in various combinations. There are several types of alternative splicing, including exon skipping, mutually exclusive exons, alternative 5′ or 3′ splicing sites, and intron retention. Alternative splicing can generate multiple mature mRNA variants from one pre-mRNA via employing different exon splicing patterns to ensure the diversity of protein isoforms in eukaryotes [17]. The dysregulation of pre-mRNA alternative splicing leads to dysfunction or absence of functional proteins, ultimately leading to diseases, including cancers and hereditary disorders [18].

### 2.3. U2AF1 Gene Structure

The U2AF1 gene (alias U2AF35), a 35 kDa protein, is a member of the serine/arginine gene family located on chromosome 21q22.3. The U2AF1 gene has three transcription variants with negative transcriptional direction. The U2AF1 protein contains four main domains, including a U2AF homology domain (UHM), a serine/arginine-rich domain (RS), and two zinc finger domains (ZnF) (Figure 2A) [16]. The U2AF1 protein (small subunit of U2AF) and U2AF2 protein (large subunit of U2AF, 65 kDa) form a U2AF heterodimer binding to the 3′SS upstream polypyrimidine tract and the AG dinucleotide site, respectively. The recognition and binding of U2AF1 to the AG dinucleotide site of the 3′SS is an essential step for the E complex formation to initiate the pre-mRNA splicing cycle (Figure 2B) [19].

## 3. U2AF1 Mutations in Myeloid Malignancies

### 3.1. Mutational Patterns of U2AF1

The precise alternative splicing regulated by splicing factors is essential for normal hematopoiesis. A decade of whole genome and exon sequencing studies in hematological tumors has identified multiple splicing factor mutations as driving factors in hematopoietic malignancies [19]. Whole-exome sequencing of paired tumor/control DNA samples from MDS patients reveals that the recurrent splicing factor mutations frequently occur in the major components of E/A splicing complexes, including U2AF1, SF3B1, SRSF2, and ZRSR2. Mutations in the components of E/A complexes may impair the accurate recognition of 3′SS and produce aberrantly spliced mRNA transcripts [21].

Numerous mutation sites of U2AF1 have been discovered in both hematologic malignancies and solid tumors, indicating the broad significance of U2AF1 mutations in tumor pathogenesis. The common missense mutations include S34F/Y, Q157P/R, A26V, R35L/C, R156H, G213A, E159, E124, E152, I24T, N38H, I46V, E184D, R198Q, and R234C. The most common mutations of U2AF1 occur in S34 and Q157 residues located in the first and second ZnF, respectively. S34F substitution (Ser34Phe, the 34th serine substituted by phenylalanine) is the most common, followed by S34Y (Ser34Tyr, the 34th serine substituted by tyrosine), Q157P (Glu157Pro, the 157th glutamate substituted by proline), and Q157R (Glu157Arg, the 157th glutamate substituted by arginine) (Figure 2C,D) [11,22].

Based on the somatic mutation data at the DNA level of 12 cancer types in the Cancer Genome Atlas database (TCGA), U2AF1 mutations are found in 7 cancer types, including AML, lung adenocarcinoma, endometrial carcinoma, bladder urothelial carcinoma, head and neck squamous cell carcinoma, colon cancer, and invasive breast carcinoma [11]. U2AF1 mutations also infrequently occur in lymphoid neoplasms and ovarian borderline mucinous tumor [12]. Remarkably, a large number of U2AF1 mutational landscapes in various hematologic tumors have been reported, including MDS, MDS/MPN, MPN, AML, and chronic myeloid leukemia (CML). Here, we summarize U2AF1 mutations in patients with hematologic malignancies detected by Sanger sequencing, next-generation sequencing, and other mutation scanning technologies from 27 studies [11,12,13,21,23,24,25,26,27,28,29,30,31,32,33,34,35,36,37,38,39,40,41,42,43,44,45]. Among them, one meta-analysis that includes 13 studies with 3038 patients and another meta-analysis that includes 14 studies with 3322 patients reported an overall incidence of the U2AF1 mutant of 11.7% in MDS patients [32,33]. U2AF1 mutation frequencies in these studies were 5–21.7% in MDS, 3.4–12.3% in AML, 0–8% in CML, 1.2–8% in MPN, and 5.7–14.5% in MDS/MPN overlap patients. These studies indicate that U2AF1 is most frequently mutated in MDS, followed by AML and MDS/MPN, but rarely mutates in CML and MPN (Table 1).

### 3.2. Correlations between U2AF1 Mutations and Clinical Features

U2AF1 mutations are early genetic events in MDS patients. U2AF1 mutations in some patients with clonal cytopenias of undetermined significance may have positive predictive value for the risk of developing myeloid neoplasms [46,47]. Thus, a full understanding of the clinical consequences of the U2AF1 mutation is necessary. Different types of U2AF1 mutations might have distinct clinical and biological consequences. Most studies revealed that MDS patients with U2AF1 mutations are younger with an age-dependent trend [27,32,38]. However, controvert data have been published that there were no significant differences in age between U2AF1^mut^ and U2AF1^wt^ patients in a cohort of 221 MDS patients [24]. Several studies have confirmed that the U2AF1 mutations happened more frequently in males than in females [24,29,33,40,44]. Specifically, patients harboring the U2AF1^S34^ mutation are predominantly male, but there is no difference in gender distribution of the U2AF1^Q157^ mutation [45]. Nevertheless, another study suggests that there is no significant gender difference between U2AF1^mut^ and U2AF1^wt^ patients [27].

Patients with MDS suffer from peripheral blood cytopenias, including refractory anemia and an increased percentage of myeloblasts in the bone marrow. In terms of peripheral blood examinations, some studies have shown that U2AF1 mutations (mainly U2AF1^S34^) are significantly associated with reduced hemoglobin level and platelet count, resulting in the higher rate of anemia and thrombocytopenia in U2AF1^mut^ MDS patients [29,36,37,38,44,48]. Moreover, lower hemoglobin and platelet levels were also observed in patients with other myeloid tumors (AML, MPN, and MDS/MPN) with the S34 mutation rather than the Q157 mutation [44,45]. A significant association between the U2AF1 mutation and anemia and thrombocytopenia has also been verified in primary myelofibrosis [49]. The mean corpuscular volume (MCV) value and myeloid to erythroid (M/E) ratio tend to be relatively low in MDS patients harboring U2AF1 mutations [28]. The U2AF1 mutation is specifically associated with trilineage morphologic dysplasia (erythroid, myeloid, and megakaryocytic) in AML with myelodysplasia-related changes [41]. However, three studies published a controvert data opinion that there was no significant association between the U2AF1 mutation and blood parameters, such as hemoglobin, platelet, and white blood cell counts, in MDS and AML patients [24,27,43]. Notably, different types of U2AF1 mutations may have different impacts on hemograms. U2AF1^S34^ MDS patients have relatively low platelet levels, whereas hemoglobin concentrations were lower in MDS patients with U2AF1^Q157/R156^, and myelofibrosis was more common [38].

Cytogenetic aberration is considered a prognostic biomarker in MDS patients. Specific gene mutations in hematological malignancies are closely related to cytogenetic abnormalities and promote the occurrence and development of the disease. Cytogenetic testing reveals that the positive cytogenetic findings are more common in U2AF1^wt^ MDS patients than in U2AF1^mut^ patients, but the U2AF1^mut^ patients had a higher proportion of poor cytogenetic findings and high-risk patients [29]. A meta-analysis found that compared to U2AF1^wt^ patients, U2AF1^mut^ MDS patients are more likely to have abnormal karyotypes [33]. Conversely, it has been suggested that U2AF1 mutations are associated with low-risk karyotypes [36]. The U2AF1 mutation is inversely correlated with complex karyotypes in MDS patients and often occurs simultaneously with an isolated +8 karyotype, suggesting a moderate prognosis. Several studies have consistently reported that the incidence of the del(20q) karyotype, which predicts a good prognosis, is higher in MDS patients with U2AF1 mutations than in patients without U2AF1 mutations [24,27,30,32,38,39,50,51]. U2AF1 mutations are enriched for del(20q) in MDS, AML, and MDS/MPN overlap patients, but not in MPN cases [44,45]. Compared to wild-type, the U2AF1^Q157P^ mutation has a higher incidence of chromosome 7 abnormalities [39].

Evaluating the molecular associations between U2AF1 and other gene mutations in patients with myeloid neoplasms can help to understand the biological significance of U2AF1 mutations. Analysis of the molecular and clinical data from 1700 MDS patients revealed the mutational spectrum of U2AF1 mutations. The top genes co-mutated with U2AF1 are mainly transcription factors and DNA methylation genes, including ASXL1, BCOR, TET2, DNMT3A, PHF6, ETV6, RUNX1, STAG2, and SETBP1. The secondary mutations of the S34 mutation include ETV6, BCOR, and CUX1, while the most common concurrent mutations with the Q157 mutation are ASXL1 and DNMT3A [23]. Several studies have demonstrated that ASXL1, DNMT3A, and RUNX1 mutations are significantly enriched in U2AF1^mut^-positive MDS cases [24,26,27,35,38,39]. SF3B1 and SRSF2 mutations are mutually exclusive with U2AF1 in patients with myeloid neoplasms [44], and SF3B1, SRSF2, and TP53 mutations are shown to be inversely associated with U2AF1 in MDS patients [27,28,35,38,39]. However, another comprehensive analysis of U2AF1 mutations in 843 patients with myeloid neoplasms holds the view that U2AF1 mutants were only positively correlated with the occurrence of ASXL1 and KIT mutants in MDS/MPN patients, but not in MDS or AML patients [44]. Moreover, there is no significant association between U2AF1 mutations and other gene mutations in a cohort of Chinese AML patients [43]. Taken together, we conclude that U2AF1 mutations mainly co-occur with ASXL1, DNMT3A, and RUNX1, but is mutually exclusive with SF3B1 and SRSF2 in MDS.

### 3.3. Impacts of U2AF1 Mutation on Prognosis and Leukemic Transformation

The effects of U2AF1 mutations on the survival and leukemia transformation of patients with myeloid malignancies remain controversial. There are four studies reporting that the presence of U2AF1 mutations has no significant impact on the overall survival (OS) or leukemia-free survival in MDS patients [26,28,33,43]. However, U2AF1 mutations/variants show a significant correlation with worse OS and progression-free survival compared to U2AF1 unmutated cases in multiple MDS patient cohorts [23,27,29,30,32,33,34,35,36,37,38,39,51,52]. U2AF1 mutations also predict poor prognosis in the young population and the low-risk subgroup of MDS patients [27]. U2AF1 mutants may even be an independent unfavorable prognostic factor in patients with de novo MDS [32]. More specifically, an allele frequency (VAF) of U2AF1 greater than 40% is an independent factor for poor OS in MDS patients [39]. MDS patients with ancestral U2AF1 mutations had a shorter OS compared with those carrying secondary U2AF1 mutations [23]. Although different types of U2AF1 mutants, including S34, Q157, and R156, have hazardous effects on survival in MDS patients, the effects of the different U2AF1 variants may differ [38]. Compared with U2AF1^S34^ mutated patients, U2AF1^Q157/R156^ mutated patients tend to have a worse prognosis in MDS patients for long-term survival [35,38,39]. However, Adema et al. drew the opposite conclusion that patients with the U2AF1^S34^ mutation have a similar OS to those carrying the U2AF1^Q157^ mutation, while patients with the ancestral U2AF1^S34^ mutation have a shorter OS than the ancestral U2AF1^Q157^ mutation [23]. Co-mutated genes of U2AF1 can also participate in affecting disease progression and prognosis. In U2AF1^mut^ MDS patients, ASXL1 and RUNX1 mutated cases may increase the risks of leukemic transformation and relapse, respectively [39]. In addition, U2AF1 mutations may not affect the response rate or survival of MDS patients who are treated with first-line hypomethylation or decitabine therapies, suggesting that U2AF1^mut^ patients can also benefit from these treatments [33,53].

Similarly, AML patients harboring the U2AF1 mutation (especially S34F) tend to have a reduced OS and relapse-free survival than those without mutations [40,43,52]. Ohgami et al. described the U2AF1 mutation as an independent prognostic factor for OS in AML patients with myelodysplasia-related changes to predict poor clinical response, OS, and disease-free survival [41]. Two independent cohorts of intermediate-risk AML confirmed that the U2AF1 mutation may act as an independent predictor of relapse and OS in patients achieving complete response and affects their clinical benefits from hematopoietic cell transplantation therapy [42]. In 2013, Damm et al. observed no association between U2AF1 mutations and the risk of progression to sAML from MDS [24]. However, more studies have shown that de novo MDS patients with recurrent U2AF1 mutations (especially S34) have an increased risk to progress to sAML and shorter time-to-leukemia transformation. However, there is insufficient evidence that the U2AF1 mutation can replace these well-accepted predictors of leukemia transformation [26,27,31,32]. Overall, the U2AF1 mutation is stable during the disease progression of myeloid tumors and may be one of the potential biomarkers for risk stratification and monitoring therapy response in patients with myeloid malignancies, especially MDS [27].

## 4. U2AF1 Mutation Affects Hematopoietic Function and Target Genes

### 4.1. U2AF1 Mutation Impairs Hematopoietic Function

Wild-type U2AF1 is critical in maintaining the survival and function of hematopoietic stem/progenitor cells (HSPCs) and normal hematopoietic development. In vitro studies demonstrated that U2AF1^S34F^-transduced hematopoietic progenitor cells show impaired hemoglobinization, increased apoptosis, impaired cell growth, and differentiation in erythroblasts. This finding may explain the relatively low hemoglobin and more anemia cases in U2AF1^mut^ MDS patients. The U2AF1^S34F^ mutation also skews granulomonocytic differentiation toward granulocytes (specifically eosinophils), which may impair the growth and differentiation of granulomonocytic cells [54]. As the most common mouse model for the mutant allele of the U2AF1 mutation, the genetically engineered mice that assemble U2af1^S34F^ generated by Cre recombinase exhibit impaired hematopoiesis with MDS-like features (multilineage cytopenia and dysplasia), as well as aberrant splicing profiles similar to the splicing alterations in U2AF1^S34F^ mutated human cells [55]. Hela and TF-1 cells transduced with mutant U2AF1^S34F^ show inhibited cell proliferation and induced apoptosis compared with U2AF1^wt^ cells. Retroviral transduction of U2af1^S34F/Q157P/Q157R^ mutants in C57BL/6 mice bone marrow impairs HSPC reconstitution, which further confirms the growth-suppressive effect of the U2AF1 mutation in vivo [21]. Compared with the U2af1^wt^ group, doxycycline-inducible U2af1^S34F^-recipient transgenic mice showed leukopenia, increased progenitor cells in bone marrow and spleen, and alterations in the distribution of mature hematopoietic lineages in bone marrow with reduced monocytes and B cells and increased neutrophils [56]. Similar to U2af1^S34F^ transgenic mice, doxycycline-induced U2af1^Q157P^ transgenic mice also exhibited a multi-lineage competitive disadvantage of bone marrow stem cells. However, there were no significant changes in peripheral blood counts and lineage distribution, indicating that these two mutations have the different effects in hematopoietic phenotypes [57]. Recently, Dutta et al. observed severe hematopoietic defects in U2af1 conditional knockout (floxed) mice, which is characterized by pancytopenia (leukocyte, neutrophil, red blood cell, and platelet), and bone marrow aplasia with reduced myeloid, erythroid, and megakaryocytic precursors. The HSPC ablation caused by U2af1 deletion may lead to bone marrow failure and early lethality in U2af1-deficient mice [58,59]. The heterozygous U2af1^mut/+^ mice generated by the CRISPR/Cas9 system exhibited aberrant RNA splicing and a defective reconstitution capacity of HSPC in transplantation assays. U2af1/Tet2 double mutant mice had increased monogranulocytic precursors, but did not succumb to MDS [60]. Briefly, these studies using different cell and mouse models strongly prove that transgenic and knock-in of the U2AF1^S34F^ mutant may lead to severe defects in HSCs and hematopoiesis.

### 4.2. U2AF1 Mutation Affects Alternative Splicing

Since the initial report of U2AF1 mutations in myeloid malignancies, a growing number of studies have focused on exploring the molecular consequences of U2AF1 mutations in cell and mouse models, and clinical specimens. U2AF1 mutants can induce RNA splicing dysfunction, including frequent exon hopping, intron retention, alternative 3′SS usage, premature stop codon, and enhanced splicing [13,20,21,22,25,31,61]. Cancer specimens bearing U2AF1^S34F/Q157R^ mutations establish new 3′SS contacts at −3 and +1 nucleotides, respectively, which impact U2AF2-RNA interactions [61]. The U2AF1^S34F/Y^ mutants show preferential enrichment of the cassette exon and alternative 3′SS, the high-frequency uridine of e-3 nucleotide at 3′SS, and a preference for CAG over UAG [11,22]. The U2AF1^Q157^ mutant reinforces the preferential recognition of G instead of A at the +1 position [62].

### 4.3. U2AF1 Mutation Alters Downstream Genes

Numerous studies have focused on uncovering the effects of U2AF1 mutations on the downstream target genes and biological pathways in myeloid cancers to uncover the driving effects of U2AF1 mutations on cancer progression. Gene Set Enrichment Analysis (GSEA) of whole transcriptome sequencing data revealed the significant down-regulated genes associated with HSC maintenance, and upregulated genes associated with the cell cycle and DNA damage response in U2af1-deficient HSPC [58]. U2af1 deficiency also leads to altered splicing and expression of the transcription factors Nfya and Pbx1 that maintain HSC survival and self-renewal [58,59]. RNA sequencing analysis of U2AF1^S34F^ transgenic mice showed altered hematopoiesis and pre-mRNA splicing in hematopoietic progenitor cells. The integrative analysis with the sequencing data from MDS patients confirmed that U2AF1^S34F^-induced splicing events are enriched in RNA splicing and processing, translation process/ribosomal genes, and recurrently mutated genes in MDS and AML [56]. Expression of the U2AF1^S34F^ mutant in mouse bone marrow stroma OP9 cells induces the release of hydrogen peroxide, cytokines, and chemokines. U2AF1^S34F^ OP9 cells exhibit mitochondrial dysfunction, oxidative stress, DNA damage, and genomic instability [63]. Nonsense-mediated mRNA decay (NMD) is an RNA surveillance process that eliminates nonsense mRNAs produced by splicing dysregulation. Compared to U2AF1^wt^ cells, U2AF1^mut^ cells are more sensitive to NMD inhibition, resulting in DNA replication obstruction, DNA damage, and chromosomal instability [64].

Dysregulated pathways associated with MDS pathology have been found in U2AF1-mutated MDS cases, including mitochondrial dysfunction, oxidative phosphorylation, heme biosynthesis, sirtuin signaling, DNA damage, and the cell cycle [25]. Two studies based on the analysis of the RNA-seq data of AML cases from TCGA found that U2AF1 mutations lead to aberrant gene splicing events in important biological pathways, such as cell cycle progression, RNA processing, and other cancer-related genes. Preliminary experiments confirmed that the U2AF1 mutation may exert biological effects by the altered oncogenes CTNNB1, CHCHD7, and PICALM [11,13]. Another study identified hundreds of differentially spliced genes caused by U2AF1 mutations from U2AF1^S34F/Y^ AML patients, U2AF1^S34F/Y^ K562, and U2AF1^Q157P/R^ K562 cells. Many of these genes have been reported to participate in the biological pathways of myeloid malignancies. Core dysregulated genes include DNMT3B in DNA methylation, H2AFY in chromosome inactivation, ATR and FANCA in the DNA damage response, and CASP8 in apoptosis [62].

In addition, some studies have deeply investigated the exact molecular mechanisms by which the U2AF1 mutation affects cell phenotypes such as ribosome genesis, autophagy, and pyroptosis. In 2017, Yip et al. reported that the U2AF1^S34F^ mutation causes impaired differentiation and an aberrant splicing process in erythroid and granulomonocytic colonies, and identified two key downstream target genes H2AFY and STRAP, which are key effectors regulating erythropoiesis and granulomonocytic differentiation [54]. Another in vivo study also confirmed H2AFY as a target of alternative splicing controlled by the U2AF1^S34F^ mutation. The U2AF1^S34F^ mutation downregulates the expression of H2afy splice isoform H2AFY1.1, and subsequently inhibits the expression of Ebf1, a key transcription factor necessary for B cell development, resulting in defective B-lymphogenesis in U2AF1^S34F^ mice [65]. U2AF1^S34F^-transformed pro-B cell line Ba/F3 shows increased use of the distal poly(A) site in Atg7 mRNA, which leads to the production of long, inefficiently translated transcripts. Decreased ATG7 protein levels result in defects in autophagy, mitochondrial dysfunction, and an increased frequency of secondary oncogenic mutations. These results demonstrate the pro-oncogenic function of the U2AF1^S34F^ mutation in driving tumorigenesis [66]. Another study demonstrates that MDS-derived AML cell line SKM-1 with the U2AF1 mutation undergoes overexpression of cancer-associated transcription factor FOXO3a and cell cycle regulators p21^Cip1^ and p27^Kip1^, resulting in proliferation inhibition, activation of apoptosis, and autophagy flux. Interestingly, FOXO3a overexpression also promotes NLRP3 inflammasome activation [52]. The U2AF1^S34F^ mutation in MDS cells facilitates the production of NLRP3 inflammasome-dependent IL-1β and IL-18 and triggers pyroptotic cell death in HSPC, a hallmark of MDS [52,67]. Interleukin-1 receptor-associated kinase 4 (IRAK4) has dominant alternatively spliced isoforms in MDS and AML patients. The U2AF1 mutation can directly mediate the production of a longer protein isoform IRAK4-Long (IRAK4-L) with oncogenic activity to mediate maximal activation of NF-κB and MAPK, and maintain the function of leukemic cells. Inhibition of IRAK4-L in AML cells with high IRAK4-L expression or the U2AF1 mutation significantly abolishes leukemic growth [68].

### 4.4. Noncanonical Functions of U2AF1 Mutation

U2AF1 also has noncanonical functions that directly regulate translation, leading to misregulation in translation initiation and ribosomal biogenesis genes. The U2AF1^S34F^ mutation can upregulate the translation level of nuclear phosphoprotein 1 (NPM1), which acts as a driving factor of myeloid tumorigenesis. The improper translation of NPM1 can lead to defective ribosomal RNA processing and impaired viability of U2AF1^S34F^ cells [69]. U2AF1 can directly bind hundreds of spliced polyadenylated mRNAs in the cytoplasm and negatively regulate mRNA translation. For instance, the U2AF1^S34F^ mutation up-regulates the translation of inflammatory factor interleukin 8 (IL8), which directly promotes oncogenic phenotypes (e.g., metastasis and inflammation) [70]. A more recent study showed that U2AF1^mut^ cell lines and patient-derived MDS/AML blasts exhibit enhanced stress granule responses and altered stress granule components [61]. These studies collectively indicate that the central biological processes targeted by U2AF1 mutations in myeloid malignancies include RNA splicing and processing, ribosome biogenesis, mitochondrial dysfunction, DNA damage, cell cycle, autophagy, and pyroptosis.

## 5. U2AF1 Mutation as a New Therapeutic Target

Given the pro-tumorigenic effects of U2AF1 mutations on leukemogenesis by causing splicing defects in specific genes, U2AF1 may be a useful target for drug discovery. A recent study showed that U2AF1 is a haplo-essential oncogene for mouse hematopoietic cancer cell survival and U2AF1 mutations are always heterozygous with the residual WT allele. U2AF1 mutant hematopoietic cells may depend on the expression of the U2AF1^WT^ allele for survival. Deletion of the U2AF1^WT^ allele in U2AF1-mutant leukemic mice could significantly reduce tumor burden and prolong survival. Therefore, given the vulnerability of hematopoietic cells with spliceosome mutations, selectively targeting the U2AF1^WT^ allele in heterozygous mutant cells has the potential to therapeutically induce cancer cell death in patients with U2AF1 mutations [71]. U2AF1 mutated hematopoietic cells, primary patient cells, and transgenic mice exhibit increased sensitivity to sudemycin treatment, a compound that regulates pre-mRNA splicing, indicating the potential of sudemycin for myeloid neoplasms patients with U2AF1 mutations [72]. Isoform modulation of U2AF1^S34F^ target genes H2AFY and STRAP can rescue the erythroid differentiation defects caused by the U2AF1 mutant in MDS cells, suggesting that splicing regulators may be effective against myeloid tumors with the U2AF1 mutation [54]. Both U2AF1^S34F^ MDS/AML cell lines and primary cells from MDS and AML U2AF1^mut^ patients are sensitive to pharmacologic inhibition of IRAK4 CA-4948. Knockdown of the target gene IRAK4-L or CA-4948 in U2AF1^S34F^ AML cells can enhance the differentiation of erythroid and myeloid cells, and abrogate leukemic growth in mice xenografted with THP1 cells [68]. In addition, autophagy deficiency can make U2AF1^S34F^-containing MDS cells highly sensitive to cytotoxic agents and autophagy inhibitors, suggesting that DNA damaging agents have therapeutic effects in U2AF1^mut^-related diseases [66].

## 6. Discussion and Perspectives

Spliceosome genes are the most common targets of mutations in patients with hematopoietic malignancies, and more than half of MDS patients have spliceosome gene mutations, revealing a new leukemogenic pathway. The U2AF1 mutant is one of the frequent molecular abnormalities in patients with myeloid malignancy, especially in MDS and AML patients with an average mutation rate around 12% and 8%, respectively. The U2AF1 mutations occur almost exclusively as S34F/Y and Q157P/R in the two zinc finger domains, resulting in RNA splicing dysfunction. Numerous reports have confirmed that U2AF1 mutations in MDS and AML patients have associations with clinical features, such as age, gender, hemograms, bone marrow, karyotype, and other mutated genes, although some conclusions remain controversial. Current studies generally agree that the U2AF1 mutation is stable during disease evolution, and may predict poor prognosis and a higher risk of leukemic transformation in MDS and AML patients (Table 1). The U2AF1 mutation is included in the somatic mutations of 16 genes in the IPSS-M and may be a potential biomarker for risk stratification and monitoring treatment response in MDS patients along with other core somatic mutations [5].

Although the precise mechanisms of U2AF1 mutants in the pathogenesis of MDS and AML remain largely unclear, a growing number of studies are dedicated to explaining the biological impacts and carcinogenic effects of U2AF1 mutations. Wild-type U2AF1 is a critical requirement for the control of HSPC survival and maintenance of normal hematopoiesis. However, many in vivo and in vitro experiments have demonstrated that U2AF1 mutations (especially U2AF1^S34F^) can severely impair hematopoietic progenitor cells and hematopoietic function. Hundreds of defectively spliced downstream targets and multiple biological pathways caused by U2AF1 mutations have been identified. These alterations help to understand the molecular mechanisms of the U2AF1 mutation in leukemia pathogenesis and develop novel therapeutic targets. Although drugs that directly target the U2AF1 mutation are currently lacking, splicing modulators targeting the U2AF1 mutation have shown potential therapeutic efficacy in U2AF1 mutated myeloid malignancies, raising the possibility of applying splicing modulators to patients harboring U2AF1 mutations. In summary, future studies need to focus on exploring the relationship between different types of U2AF1 mutations and the clinical characteristics and prognosis of MDS and AML patients to more accurately guide disease risk stratification and prognosis assessment. In addition, future research should explore the exact molecular mechanisms of U2AF1 mutations in driving leukemogenesis to improve the precision treatment for myeloid malignancies with U2AF1 mutants.

## Figures and Tables

**Figure 1 cancers-14-04406-f001:**
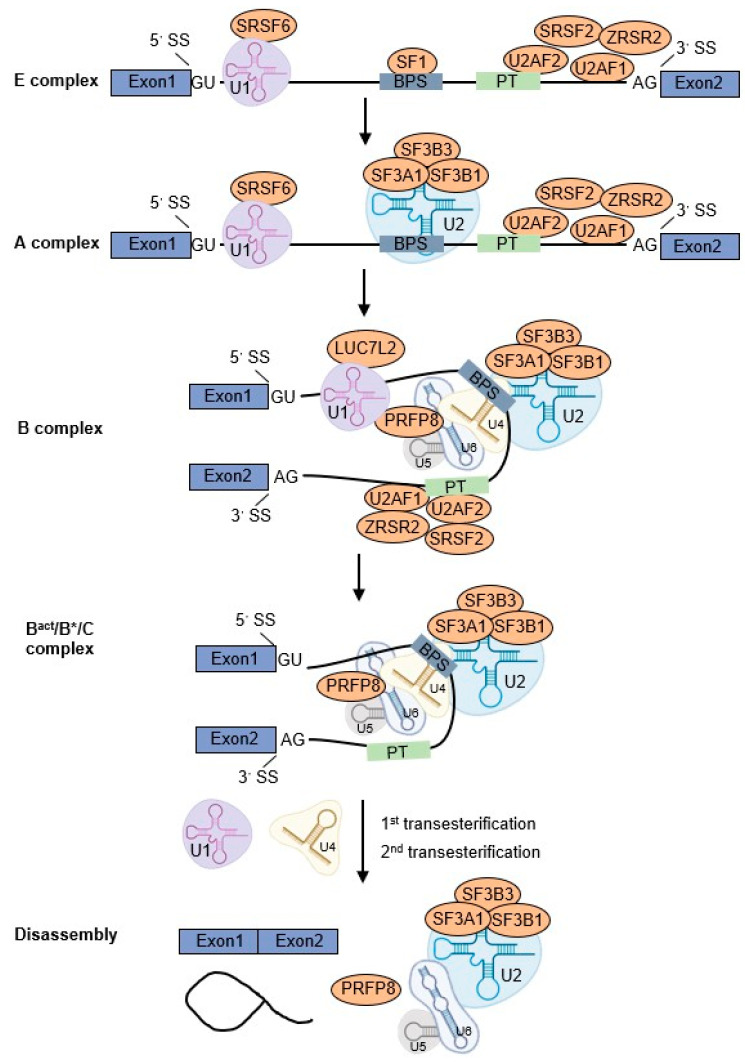
U2AF1 participates in the pre-mRNA splicing process. The splicing process of U2-dependent spliceosome is the orderly recruitment of U1, U2, and U4/U6.U5 tri-snRNP complexes to bind to the 5′ splicing site (5′SS) or branch-point sequence (BPS) to form E, A, B, and Bact/B*/C complexes successively under the participation of multiple splicing factors. U2AF1 binds to the AG dinucleotide site of the 3′SS and collaborates with other splicing factors (U2AF2, SRSF2, and ZRSR2) to recognize the 3′SS and recruit U2 snRNP to bind the BPS. Finally, two transesterification reactions are catalyzed to form and release an intron loop/lariat, and join the exons to synthesize mature mRNA.

**Figure 2 cancers-14-04406-f002:**
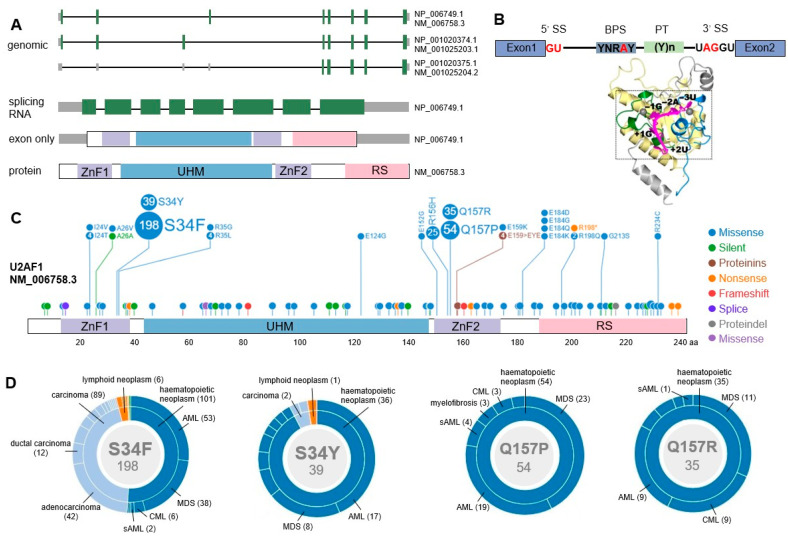
Gene structure and mutations of U2AF1. (**A**) Schematic representation of three transcript variants of U2AF1, and the splicing RNA and protein of isoform a. Green boxes represent exons. ZnF: zinc finger domain, UHM: U2AF homology domain, RS: serine/arginine-rich domain. NM_006758.3→NP_006749.1 (isoform a), NM_001025203.1→NP_001020374.1 (isoform b), NM_001025204.2→NP_001020375.1 (isoform c). (https://pecan.stjude.cloud/proteinpaint/U2AF1, accessed on 1 August 2022). (**B**) Schematic diagram of U2AF1 protein binding to RNA at the 3’ splicing site (3′SS, 5′-UAGGU). PT: polypyrimidine tract, Y: pyrimidine, R: purine, N: any nucleotide. Adapted from Yoshida H et al. [20]. (**C**) All mutation sites of U2AF1 protein (isoform a). The mutation profile is visualized using the mutation data from COSMIC database embedded in ProteinPaint. Circles in different colors represent different types of mutations, and the numbers in the circles indicate the detected mutation numbers in clinical specimens. (**D**) The distributions of the four major mutations of U2AF1 (S34F/Y, Q157P/R) in cancers. The diseases and their corresponding numbers of U2AF1^mut^ cases are labeled. The inner circle represents disease categories, and the outer circle indicates more disease subtypes. Due to the space limitations, only U2AF1 mutations in hematological malignancies are labeled in the figure. Complete mutation information can be found on the above website.

**Table 1 cancers-14-04406-t001:** The types and clinical implications of U2AF1 mutations in myeloid neoplasms.

Disease (Frequency)	Mutation Types	Sample	Clinical Consequences	Ref.
MDS (5%)	S34, Q157, Q84, E124, E152, R156	Targeted deep sequencing of 1700 patients with myeloid neoplasms	Ancestral U2AF1 mutations predict shorter OS compared to secondary mutations; S34 co-occurs with ETV6, BCOR, and CUX1; Q157 co-occurs with ASXL1 and DNMT3A	[23]
MDS (5.4%)	S34F/Y, Q157R/P/fs	Mutation analysis of 221 MDS patients	Associated with chromosome 20 deletions and ASXL1 mutation; Not associated with the risk of progression to sAML	[24]
MDS (7.1%)	S34, Q157, R156	Targeted next-generation sequencing of 84 MDS patients	Associated with a high proportion of exon skipping and retained introns events	[25]
MDS (7.3%)	S34F, Q157P, E159fs	Sequencing analysis of 193 MDS patients	Positively correlated with ASXL1 and DNMT3A mutation; Not associated with the presence of ring sideroblasts; Have no impact on patient survival; Have a trend toward a more rapid progression to AML; (Patients received treatments including all-trans retinoic acid, antithymocyte globulin, deferasirox, lenalidomide, or thalidomide)	[26]
MDS (7.5%)	S34F/Y, Q157R/P	Direct sequencing in 478 patients with de novo MDS	Positively correlated with isolated −20/20q-, ASXL1, RUNX1, and DNMT3A mutations, negatively correlated with SRSF2 mutation; Independent poor-risk factor for OS in MDS patients; Predict shorter time-to-leukemia transformation	[27]
MDS (7.5%)	S34F/Y, Q157P	Sanger sequencing of 106 MDS patients	Associated with low mean corpuscular volume and myeloid to erythroid ratio; Have no impact on OS	[28]
MDS (7.8%)	S34F/Y, Q157P	Mutation analyses of 129 de novo MDS patients without ring sideroblasts	Associated with low hemoglobin levels and high-risk MDS; Predict poor PFS; Associated with inferior OS in low-risk MDS patients	[29]
MDS (8.6%)	S34F/Y, Q157P	Next-generation sequencing of 304 Chinese MDS patients	More common in patients with trisomy 8 or 20q deletions; Predict poor OS in MDS patients	[30]
MDS (8.7%)	S34F/Y	Sanger sequencing of 150 patients with de novo MDS	Enhance splicing and exon skipping; Increased risk of progression to sAML	[31]
MDS (5–17%, 11.7%)	S34, Q157, R156	Meta-analysis of 14 studies with 3322 MDS patients	Independent, detrimental prognostic factors for OS and AML transformation	[32]
MDS (11.7%)	S34, Q157	Meta-analysis of 13 studies with 3038 MDS patients	Associated with poor OS, but not DFS; Q157 mutation predicts worse OS than S34; Have no impact on hypomethylating therapy	[33]
MDS (14%)	-	Targeted capture assays of 300 primary MDS patients	More prevalent in the intermediate-risk cytogenetic category; Unfavorable survival impact	[34]
MDS (15%)	S34F/Y, Q157R/P, R156H	Next-generation sequencing of 357 primary MDS patients	Not associated with anemia; Q157 positively correlated with ASXL1 mutation; Q157 mutation has adverse survival impact	[35]
MDS (16%)	-	Next-generation sequencing of 179 primary MDS patients	Associated with lower-risk karyotype and platelet count; Adverse survival impact	[36]
MDS with lower risk (16%)	-	DNA sequencing of 288 patients with MDS	Associated with low platelet count and shorter overall survival	[37]
MDS (17%)	S34F/Y, Q157R/P, R156H	Targeted gene sequencing of 511 MDS patients	Associated with anemia, thrombocytopenia, ASXL1 mutation, isolated +8, and poor survival; Inversely associated with TP53, SF3B1 mutations, and complex karyotypes	[38]
MDS (21.7%)	S34F/Y, Q157P	Retrospective analysis of the next-generation sequencing data of 234 MDS patients	Positively correlated with ASXL1, RUNX1, and SETBP1 mutation; negatively correlated with SF3B1 and NPM1 mutation; VAF > 40% of U2AF1 is an independent indicator for poor OS of MDS patients	[39]
AML (3.4%)	S34F/Y, R35Q	Nanopore sequencing of 1119 AML patients	Predict poor OS in AML patients	[40]
AML (4%)	S34F/Y	Somatic mutation data from TCGA AML patients	Preferentially exhibit alterations in cassette exon and alternative 3′SS; Preferentially splices to CAG rather than UAG	[11]
AML (6.5%)	-	Targeted next-generation sequencing of 93 AML patients	Associated with AML with myelodysplasia-related changes and trilineage morphologic dysplasia; Associated with the absence of clinical remission, poor OS and DFS	[41]
AML (11%)	-	Targeted sequencing in 100 intermediate-risk AML patients	Predict poor OS and RFS	[42]
MDS (19.7%) sAML (4.6%)	S34F/Y	DNA sequencing of 2345 tumor tissues	-	[12]
MDS (6.3%) AML (2.5%) CML (0%)	S34F/Y, Q157R/P	Mutation scanning of 275 primary AML, 96 primary MDS, and 81 CML Chinese patients	Predict poor OS, but not DFS, in AML patients; Have no impact on OS in MDS patients	[43]
MDS without RS (11.6%) CML (8%) sAML (9.7%) AML (1.3%) MPN (1.9%)	S34F/Y, Q157R/P, A26V	Whole-exome sequencing of paired tumor/control DNA from 29 patients with myelodysplasia	Suppress cell proliferation and induce apoptosis; Induce abnormal RNA splicing and compromised hematopoiesis	[21]
MDS (10%) sAML and AML (7%) MPN (8%) MDS/MPN (14.5%) MDS with high risk (14%)	S34F/Y, Q157R/P, A26V, R35L, R156Q, G213A	Sanger sequencing and exome sequencing of 524 patients with hematologic malignancies	Associated with exon skipping; Induce abnormal splicing of genes in important pathways	[13]
MDS (10.9%) AML (9.5%) MDS/MPN (7.1%) MPN (1.2%)	S34, Q157	Melting curve analyses or next-generation sequencing of 843 patients	Associated with lower hemoglobin levels and platelet counts; Associated with del(20q) in MDS, AML, and MDS/MPN	[44]
MDS (S34 14.6%, Q157 1.1%) AML (S34 12.3%, Q157 0%) MDS/MPN (S34 2.2%, Q157 3.5%) MPN (S34 0.6%, Q157 0.6%)	S34, Q157	Melting curve analysis of 785 patients	S34 mutation associated with low hemoglobin level and platelet count; Associated with del(20q) in MDS	[45]

Abbreviation: OS: overall survival; PFS: progression-free survival; DFS: disease-free survival; RFS: relapse-free survival; VAF: allele frequency; TCGA: the Cancer Genome Atlas; 3′SS: 3′ splicing site.

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
