# Peer review of "The Biological and Clinical Consequences of RNA Splicing Factor U2AF1 Mutation in Myeloid Malignancies"

_cancers, 2022, doi:10.3390/cancers14184406_

Round 1

Reviewer 1 Report

Minor comments:

-          When describing animal models, the authors do not quote the following studies also relevant for the field: a) Alberti et al. Mutant U2AF1S34F and U2AF1Q157P Induce Distinct RNA Splicing and Hematopoietic Phenotypes In Vivo. Blood. 2019;134(Supplement_1):770-770. doi:10.1182/blood-2019-123892 (about Q157 mutation in vivo); b) Martinez-Valiente C et al. Int. J Mol Sci. 2021 (about U2AF1/TET-2 co-mutation, as described in lines 238-240); c) Wadugu et al. Journal of Clinical Investigation. 2021. It is recommended to include the conclusions raised in these studies in this review.

-          Line 18: a space is missing ((MDS)and)

-          Line 63: correct "extons" to exons

-          Line 122: Eliminate the word server

-          Line 125-126: This sentence is incomplete and misleading

-          Table 1: Reference 31 is missing in the table.

-          Table 1: In reference 33, correct "mutataion" to mutation

-          Line 255: change "reported" to reporting

-          Line 320: change "utilized" to using

-          Line 335: change "Numbers of studies" to numerous studies

-          It would be clearer if the paragrah describing non canonical functions of U2af1 (translation initiation and stress granules) would constitute another section (point 4.4)

-          Line 255: change "reported" to reporting

-          Line 340: add reference on GSEA analysis.

-          Line 419: add "cells":  …..the differentiation of erythroid and myeloid CELLS.

Reviewer 2 Report

This review paper is well written that has the potential to be accepted, but some points have to be clarified or fixed before publishing this paper. You will find my comments below.

1) Page 8, line 189. Is the following phrase, “An early genetic event in MDS patients” means the role of U2AF1 mutation in clonal ICUS (CCUS)? The authors may have missed a critical reference (Blood. 2017; 129: 3371-8).

2) Page 8, lines 189-190. The authors also mentioned that “a full understanding of the clinical consequences of U2AF1 mutation is necessary”. The authors should mention not only this mutation status difference in age and gender but also the differences due to race or ethnic groups.

3) Page 9, line 218. Related to the previous comment: Recently, Alaggio R et al. presented an overview of the upcoming 5th edition of the WHO Classification (Leukemia. 2022; 36(7): 1720-48). Hypoplastic MDS (h-MDS) is listed as a distinct MDS type in this edition. How about the U2AF1 mutation status in h-MDS? There is no description regarding this point in the manuscript.

4) Page 10, line 255. The authors cited four reports on the relationships between the presence of U2AF1 mutations and the prognosis in MDS patients (Reference #24, 26, 31, 41). The authors should be clarified the treatment status or options (received HMAs, HSCT, or best supportive care?) among these patients.

5) Page 10, line 285. Secondary acute myeloid leukemia (sAML) includes therapy-related AML and AML evolving from an antecedent hematological disorder not only MDS but also other relative diseases like aplastic anemia. Reference #22 covers MDS only. The authors should add the term “from MDS” at the end of the sentence.

6) There is a typo “extons” on Page 2 line 63. In addition, symbols for genes should be italicized in general.

Reviewer 3 Report

Major comments

·         Please review the largest series of MDS, AML and other myeloid malignancies where mutational landscape was described. It is true that U2AF1 mutations are among recurrent events but they are not among the most frequent molecular abnormalities as it is mentioned in the discussion.

·         Recently, the Molecular International Prognostic Scoring System (IPSS-M) for  Myelodysplastic Syndromes was published (Bernard et al. NEJM. 2022). U2AF1 is included among the sixteen genes that were found to be relevant for the IPSS-M model. Due to the relevance and novelty of such publication, authors could comment on this.

Please consider IPSS-M in the discussion section when it is mentioned that U2AF1 is not included in prognosis scoring systems.

·         Table 1: denotes that an exhaustive literature revision was made, however a more interesting, and easier to interpret table can be made based on the data described in the table. Eg: to condense the data related to each disease subtype instead of doing it based on the reviewed article.

·         Correlations between U2AF1 mutation and clinical features: it is true that controvert results are published regarding U2AF1 mutations and their associations with clinical features. The authors have reviewed several publications, however opposite data can be explained in a different way along the review. For example: “most studies reveal….. “… “controvert data has been published regarding XXXX, however, it is likely that …”. Data could be summarized in a more comprehensive way.

·         Line 225: “U2AF1mut MDS 224 patients have an increased risk of having abnormal karyotypes”… too strong statement. In the meta-analysis of Li et al. it is mentioned the following: “Abnormal karyotypes were encountered significantly more often in patients with than without U2AF1 mutations (49.6% vs. 37.8%, P < 0.05); similarly, patients with abnormal karyotypes were significantly more likely to have U2AF1 mutations (19.3% vs 12.1%, P < 0.05)”. This means that an association was found between both features, but not necessarily a risk.

·         Figure 2: please restructure the footnote. Check for spelling and citations. Please consider to redesign figure 2D: a more detailed explanation of colours, and numbers should be provided for a better understanding. The sum of the numbers of each pathology does not match with the highest number.

·         Line 290, where authors mention “Overall, U2AF1 mutation is stable during the disease progression of myeloid tumors and may be a potential biomarker for risk stratification and monitoring therapy response in patients with myeloid malignancies, especially MDS”… This statement is based in a single study that used sanger sequencing in a cohort of MDS patients. According to more recent publications (such as the IPSS-M published by Bernard et al), there are several myeloid related genes that could be used for risk stratification, rather than just one single gene.

·         When authors comment on U2AF1 as a new therapeutic target, they mention the pharmacologic inhibition by CA-4948 compound. Was it tested on animal models or patient cells? Maybe a more detailed explanation on in can be provided.

Minor comments

 ·         Please consider italics to denote gene name (see HUGO guidelines in Bruford et al., 2020.)

·         Careful reading should be made to check for spelling, spacing and punctuation errors. Eg: Line 18: space is missing after “(MDS)”. Line 63: check spelling. Line 150: delete extra space. Line 255, 440-442: check for grammar. Line 303: check for word order, “most common mouse model” or “mouse model for most common U2AF1 mutation”? Line 320: word missing

Round 2

Reviewer 3 Report

Even when not all the suggestions were considered, this version has significantly improved. 

Please read the manuscript carefully one last time and check for spelling, spaces and punctuation.